# Assessment of a Passive Lumbar Exoskeleton in Material Manual Handling Tasks under Laboratory Conditions

**DOI:** 10.3390/s22114060

**Published:** 2022-05-27

**Authors:** Sofía Iranzo, Alicia Piedrabuena, Fernando García-Torres, Jose Luis Martinez-de-Juan, Gema Prats-Boluda, Mercedes Sanchis, Juan-Manuel Belda-Lois

**Affiliations:** 1Instituto de Biomecánica de Valencia, Universitat Politècnica de València, 46022 Valencia, Spain; sofia.iranzo@ibv.org (S.I.); alicia.piedrabuena@ibv.org (A.P.); fernando.garcia@ibv.org (F.G.-T.); mercedes.sanchis@ibv.org (M.S.); 2Centro de Investigación e Innovación en Bioingeniería (Ci2B), Universitat Politècnica de València, 46022 Valencia, Spain; jlmartinez@eln.upv.es (J.L.M.-d.-J.); gprats@ci2b.upv.es (G.P.-B.)

**Keywords:** exoskeleton, lumbar, EMG, motion-tracking, fatigue, manual material handling

## Abstract

Manual material handling tasks in industry cause work-related musculoskeletal disorders. Exoskeletons are being introduced to reduce the risk of musculoskeletal injuries. This study investigated the effect of using a passive lumbar exoskeleton in terms of moderate ergonomic risk. Eight participants were monitored by electromyogram (EMG) and motion capture (MoCap) while performing tasks with and without the lumbar exoskeleton. The results showed a significant reduction in the root mean square (VRMS) for all muscles tracked: erector spinae (8%), semitendinosus (14%), gluteus (5%), and quadriceps (10.2%). The classic fatigue parameters showed a significant reduction in the case of the semitendinosus: 1.7% zero-crossing rate, 0.9% mean frequency, and 1.12% median frequency. In addition, the logarithm of the normalized Dimitrov’s index showed reductions of 11.5, 8, and 14% in erector spinae, semitendinosus, and gluteus, respectively. The calculation of range of motion in the relevant joints demonstrated significant differences, but in almost all cases, the differences were smaller than 10%. The findings of the study indicate that the passive exoskeleton reduces muscle activity and introduces some changes of strategies for motion. Thus, EMG and MoCap appear to be appropriate measurements for designing an exoskeleton assessment procedure.

## 1. Introduction

The health and comfort of industry workers are important issues, and making workstations safer is a priority. Musculoskeletal disorders (MSDs) continue to be the main cause of non-fatal injuries that require work leaves. Specifically, back injuries, including those of the spine and spinal cord, account for 17% of all injuries (2015) [1]. Reducing these numbers endorse the efforts to design more ergonomic workplaces.

The desire to reduce such numbers encourages efforts to design more ergonomic workplaces. Many technological improvements have been implemented for industry workstations, such as those in warehouses, and the physical demand of certain tasks has been substantially reduced. Further, ergonomics specialists have contributed to reducing the incidence of workplace injuries and their consequences by implementing ergonomics criteria for workstations [2,3,4].

However, there are procedures that are still required to be done manually. Many tasks in those kinds of workplaces require individuals to perform in poor working conditions to meet task demands, especially tasks that involve handling weights and pulling or pushing objects [5,6]. Repetitive back bending and twisting while handling weights can have a significant impact on postural stress [7,8]. Studies have reported that operators whose workplaces involve lifting, pulling, and pushing tasks are significantly more likely to suffer from lower back injuries compared to other kinds of workers without exposure to such handling tasks [6,9,10,11].

In the last years, exoskeletons have emerged as support devices to help reduce the risk of MSDs at jobs where a better design of the workplace or automation of procedures is not possible [12,13,14]. The different types of exoskeleton can be divided into active, which need power sources to work [15,16], and passive, which are based on a system of mechanisms and springs [17]. Passive exoskeletons are lighter and less complex than the active ones. They are often chosen to be implemented in workstations in the industry where load handling occurs, and also, were chosen as an object of the present study.

The design of the exoskeleton is focused on which body part they are intended to relieve while doing a particular kind of task. There are, among others, upper limb exoskeleton for helping in tasks performed overhead [18]. In addition, as the case of this study, exoskeletons developed to protect the low back muscles in tasks that involve manual material handling [16,19].

There are many brands that are already commercializing passive lumbar support devices, with designs that apply continuous torque to assist the lumbar joint by transferring part of the effort to the limbs [20,21]. In the present work, a widely used passive exoskeleton is chosen as the object of study [17,19,22,23,24].

The discussion regarding the objective effect of exoskeletons in terms of reducing the risk of MSDs is still on the table [25]. Many researchers are conducting tests to quantify the benefits of devices worn on the body, although no standard for evaluation practice is defined yet [26]. In particular, studies assessing lumbar exoskeleton are attempting to quantify the differences in conditions with and without an exoskeleton while performing weight handling tasks [20,27,28].

Furthermore, the validation of the effects of the exoskeleton over the fatigue reduction is still an open object of study. Some of the studies carried out in this topic will be reviewed with the aim to achieve a deeper understanding of the effects of a lumbar exoskeleton in a simulated pick-and-place workplace in terms of fatigue. Fatigue analysis supposes a step beyond previous works of the present group about exoskeleton assessment [25].

First of all, to assess the exoskeleton in the appropriate physical demanding conditions, designing a series of tasks was fundamental. The criteria followed in the present work were based on the definition of ergonomic risk levels given by literature [29,30]. These criteria use a group of factors, including weight, symmetry, distance, and frequency, of load handling to define a risk index, the composite index (CI), which is related to the probability of injury in the dorsolumbar area. With the designed tasks, posture and weight were controlled in both conditions, with and without exoskeleton, in order to gain a better understanding of the mechanical and physiological support that the exoskeleton provides to the worker.

Secondly, to deeply investigate muscle fatigue, analysis of muscular activation patterns was performed by working out activation parameters, together with parameters specifically associated with the fatigue process. The parameters were obtained under controlled postures, and their evolution was calculated throughout the development of the tasks. This study adds new knowledge to previous studies in the field, providing a deeper view of muscle behavior in terms of fatigue.

Lastly, to fulfill the main objective, motion was also captured. These data had a double objective, firstly, they allowed tracking the body position over time with the objective of completing the EMG signal tagging and segmentation. Secondly, performing a simple analysis of the ranges of motion that was included in the present work. The main purpose of this last analysis was to check that no major restrictions in the coordinates related to the studied muscles were found, to discard the possible influence in the fatigue.

With the results of this study, it is possible to conclude that the evaluated lumbar exoskeleton has an effect on the user in respect to muscle activity, fatigue, and freedom of movement. Regarding muscle activity, the effect was beneficial in all cases and no damage was found. The benefits were seen in reduced muscle activity and the fatigue process. In the case of motion, constrictive changes were found, implying a drawback. A second conclusion was that measuring and analysing muscle activity and motion are appropriate in order to develop a protocol to assess the effect of exoskeletons.

## 2. State of the Art

### 2.1. Passive Exoskeletons and Common Evaluation

There are numerous studies focusing on the passive exoskeletons effects evaluation using different methodologies. Pesenti et al. [26] classifies the types of quantification of the exoskeleton effects in five domains of criteria: functional, force and/or torque, metabolic, subjective, and muscular.

Concerning the functional domain, studies include kinematics, which is measured through motion capture in order to estimate the postural changes that the device might introduce that could have potential negative side effects, such as discomfort or a collateral risk of injury [18,31]. In addition, some authors Simon et al. [32] assessed changes in joint angle ranges, together with changes in the position of body parts, and changes in the velocity while performing freestyle tasks.

Besides, joint moments and loads are measured in order to evaluate how joints effort varies under the conditions with and without exoskeleton. Koopman et al. [22] quantifies the contribution of the exoskeleton to the L5/S1 net moment, and, Picchiotti et al. [33] studies the contribution to the moments occurring in lumbar spine while lifting weights.

Other type of assessment is performed through the evaluation of metabolic cost under the conditions with and without exoskeleton [21,31]. The subjective assessment remains one of the criteria for assessing exoskeletons in terms of perceived discomfort by the authors Luger et al. [34] and Amandels et al. [19]. However, also, in terms such as perceived freedom of movement, perceived general fatigue, or ease of use in other studies [20,23].

Muscular analysis, through the use of electromyography (EMG) is the preferred measurement method in studies of exoskeleton assessment. In many recent studies, the activation of some of the muscles involved was measured while the participants performed a series of repetitive weight lifting tasks [22,35]. In the study of Antwi-Afari et al. [27], the signals collected were differentiated by lifting or loading the weights; in the data analysis, they compared the values of mean activation when handling different weights finding significant reductions when using the device. In other studies the EMG measurement was performed in real working conditions and calculated the differences [19]. This study, also obtained significant differences, reduction in the activity of the objective muscle, and an increase of the activity of other muscles. In real conditions studies, the studied motions are realistic, but the comparison between conditions is less controlled. In some studies, the muscle activity was measured to introduce the data into a electromyography-driven spine model [19,33,35] that could allow other calculation such as tensions in certain joints.

Most of the studies where the EMG was analyzed, focused on assessing the effects of the exoskeleton by calculating the differences in amplitude of muscle activity. Apart from these outcomes that are of great relevance, EMG also could give information about the process of muscle fatigue [36]. Studies have reported that muscle fatigue can be assessed by surface EMG, noting that electrophysiological fatigue, as assessed by EMG, precedes mechanical fatigue. Fatigue is reflected in the EMG signal as increased amplitude and a spectral shift to lower frequencies [37]. The EMG amplitude is increased due to the larger amount of cells recruited to maintain the force exerted. The displacement of spectral content toward low frequencies may be caused by an increase in the duration of the motor unit action potential and the consequent decrease in muscle fiber conduction velocity [38].

### 2.2. Reduction of Fatigue Assessment

As mentioned in the Introduction, the device that is object of study is a passive lumbar exoskeleton. The effect of these devices over muscle fatigue has led to no clear consensus this far. However, some studies can be found in literature that addressed the problem of quantifying the fatigue through different approaches.

The work of Poon et al. [39] aimed to asses this problem through the study of muscle activation, lifting endurance, and oxygen consumption. The analysis carried out with the EMG data was the obtain of several percentiles of the signal Root-Mean-Square (RMS). Their results leaded to an increase of the endurance and a decrease of the lumbar activity when wearing the exoskeleton. These outcomes concerning endurance were related by the authors to a reduction of fatigue. Bosch et al. [35] also reported three times increase of endurance time when using a passive lumbar exoskeleton.

The work of Lotz et al. [40] characterized the impact over fatigue by three aspects. Firstly, as the previous authors, with the evaluation of muscular endurance; also, with the evaluation of the perceived exertion from the participants. Lastly, by the calculation of the minimal median frequency from the frequency domain of the EMG signals. This last approach, was also shared by Godwin et al. [41], who calculated the median frequency to evaluate the fatigue. In both cases, the median frequency presented significant differences between the two conditions. The latest authors, also attributed a reduction of fatigue to the reduction of the torque in the lumbar extension. In addition, Dewi and Komatsuzaki [42] gathered subjective feedback of fatigue from the participants, but the results were varied and there was not agreement.

As well as existing a great diversity of exoskeleton designs, types of workstation and tasks that compound them; also, there are differences on the approaches to value exoskeletons performance by the literature. For this reason, additional studies are still needed in order to bring new conclusions on the evaluation of the effects over fatigue introduced by these assistive devices. Not only in the objective muscles, but also, in the muscles that could potentially absorb an extra load from the ones released. In the present work, the approach followed to assess the fatigue is focused on the analysis in time and frequency domains of the EMG signals.

### 2.3. Fatigue Assessment by Temporal and Spectral EMG Parameters

Muscle fatigue is considered to result in a reduction in force generation capacity. The underlying mechanisms of this phenomenon seem to be multifactorial. Apart from psychological issue, neuromuscular features referable to the central and peripheral nervous system can be involved in fatigue phenomena and hence central and peripheral fatigue can be distinguished. The central factors of fatigue comprise decreases in the voluntary activation of the muscle, which is due to decreases in the number of recruited motor units and their discharge rate. Peripheral factors of muscle fatigue include alterations in neuromuscular transmission and muscle action potential propagation and decreases in the contractile strength of the muscle fibers. Therefore, several aspects are involved in the fatigue process (reduction of conduction speed, cell firing rate, number of cells recruited or morphology modification of the action potential) that can differently affect depending on the type of exercise to be performed (static or dynamic muscle contraction exercises) [43].

There is much research work regarding muscular fatigue assessment, by defining a set of temporal and spectral sEMG features. This is because of the fact that the sensitivity and robustness to detect fatigue of each sEMG parameter changes depending on the physiological factors involved in each fatigue process.

For instance the amplitude of sEMG signals is influenced by the number of active motor units, their discharge rates, and the shape and propagation velocity of the intracellular action potentials and it is affected by a lengthening of intracellular action potentials (IAP) [44]. However, sEMG amplitude also depends on the distance between the recording position and the fibers: if the electrodes are placed close to the active fibers sEMG signal could decrease because of fatigue but may stay practically unaffected when muscular fibers are at farther distances.

Zero crossing expresses fatigue from sEMG data, considering that activated muscle cells will generate an increment in action potentials and an increment in zero crossings [45]. Zero crossing seems to be very sensitive to fatigue onset, with noticeable decrease caused by a reduction in conduction of electrical current in the muscle [45].

Regarding spectral parameters, mean and median frequencies are related to changes in muscle fiber conduction velocities and consequent variations in the duration of the motor unit action potential. Nevertheless controversial results were obtained between static and dynamic fatigue tasks: during static contractions, the mean frequency usually decreases and some authors found decrements of the mean power frequency during dynamic fatiguing tasks, whereas others observed no change during walking exercises [46]. This contradictory behaviour can be somewhat attributed to other factors that cause changes in the EMG spectrum such as intramuscular temperature, which can induce a shifting in the sEMG signal spectrum toward higher frequencies or increasing the mean frequency. Both effects can compensate for each other, and the reductions found in the mean frequency can be not noticeable.

To overcome this issue Dimitrov et al. [36] proposed an spectral index (Dimitrov’s spectral index) based on previous indices of peripheral muscle fatigue, computing ratios between sEMG power spectral density content in high and low frequency bands. These moments are quantitative measurements of the shape of a signal.Dimitrov et al. [36] suggested that ratios between different spectral moments calculated over the power spectral density that were obtained using the discrete Fourier transform, achieved higher sensitivity under both isometric and dynamic contractions than conventional parameters (mean and median frequency) [36]. Specifically, ratios of moments of order 1 and moments of order 2 and higher were proposed because they emphasize the increase in low and ultralow frequencies in the sEMG spectrum due to increased negative after-potentials during fatigue. The best results were obtained for the spectral index that made use of order 5 [36].

## 3. Materials and Methods

The inclusion requirements were as follows: working age, 30 to 45 years old, body mass index (BMI) between 18.5 and 25.5 kg/m${}^{2}$, and in good physical shape; we also aimed to have a gender-balanced sample. The exclusion factors were the presence or history of musculoskeletal lesions or respiratory or cardiovascular pathologies. A total of 8 volunteers (4 women and 4 men) took part in the study. The participants came to the IBV facilities and provided written consent for the use and publication of their data for the purposes of this study. The average and standard deviation of the ages, weights, and heights of the participants were 35±5 years, 67.9±7.8 kg and, 175.6±4.6 cm, respectively.

### 3.1. Measurements Protocol and Setup Design

The design of tasks was carried out to emulate a common task in industry and warehouses, where the physical load for manual handling is usually very high, and to follow a series of ergonomics requirements. While these tasks do not represent all of the possible postures in the process of carrying heavy objects, the task designed is a typical depalletizing job with musculoskeletal risks due to forced postures. The chosen tasks reproduce a workstation that has little dynamic movements and that could require lower assistance. The motivation is that these jobs are the ones that passive exoskeletons appear to be more suitable for. If tasks are too heavy and dynamic would demand more complex active exoskeletons [47,48].

One of the main starting premises when designing the task was the level of risk. This is characterized by the composite index (CI) [29,30,49], which expresses risk associated with manual material handling with significant changes in handling conditions, and determines the probability of injury in the dorsolumbar region. For the purposes of this study, it should be moderate, which corresponds to a CI value between 1.0 and 1.6 for the general population [29]. To design tasks with such conditions, Ergo/IBV ergonomic risk assessment software was used [50]. The software is based on procedures for evaluating ergonomic risks in manual materials handling tasks validated in bibliography [51,52,53,54]. This software calculates the CI of work stations by characterizing them using a list of factors. The basic design of the tasks, which were the starting point, consisted of a set of repetitions of multiple-load manual handling (with significant changes in some of the variables associated with weight handling). Using the Ergo/IBV, the tasks were planned to consist of three series of depalletizing pallets with 4 rows of 4 boxes of 7, 8, and 9 kg in each series (Figure 1). The series covered moderate CI values in a wide range, from close to 1 to very close to 1.6.

The variables introduced in the Ergo/IBV in order to achieve such CI values are as follows:Duration of the task: Short, less than 1 hour of manipulation, fulfilling the recovery period of 1.2 times the work period.Frequency: An average frequency of 10 boxes lifted per minute was established, which means a lift every 6 s.Horizontal location: Position at the destination remains constant (25 cm), whereas, position at the origin varies according to the configuration of the boxes in the pallet (30 to 62 cm).Vertical location: As in the case of horizontal position, position at the destination remained constant (75 cm) and at the origin varied (29 to 104 cm).Coupling: The grip is considered to be good. The user holds the box with both hands, the grip is comfortable, the boxes have handles, and there are no improper hand/wrist postures when handling the boxes.Angle of asymmetry: An asymmetry angle of 45° is set when the user takes the boxes from the pallet (origin) and there is no asymmetry when the boxes are placed on the table (destination).

The user takes the boxes from the pallet following a previously established pattern, to ensure that all users perform the task in the same way. The 16 boxes are numbered following the order illustrated in Figure 1 (left):First row (top), boxes 1–4.Second row, boxes 5–8.Third row, boxes 9–12.Fourth row (bottom), boxes 13–16.

In total, the users performed the depalletizing task six times. The exoskeleton condition and the box weight of each repetition are as follows:First: 7 kg without exoskeleton.Second: 8 kg without exoskeleton.Third: 9 kg without exoskeleton.Fourth: 7 kg with exoskeleton.Fifth: 8 kg with exoskeleton.Sixth: 9 kg with exoskeleton.

The order followed for wearing the exoskeleton ensures the worst case possible, for a higher presence of fatigue is present at last three repetitions. In addition, the technician indicates the rhythm, every 6 s (frequency of manipulation), to the subject by using a metronome.

### 3.2. Equipment

EMG data were measured using a Noraxon wireless electromyography system (Ultium^TM^ EMG) with 4 channels monitoring the myoelectric activity of 4 muscles: erector spinae, gluteus medius, quadriceps femoris, and semitendinosus. The sampled frequency was 2000 Hz and was previewed and exported using the Noraxon myoMUSCLE^TM^ software. The bipolar electrodes were located according to the SENIAM guidelines [55]. The chosen muscles were all measured at the left side of the body, due to the asymmetric characteristics of the tasks designed.

The Xsens^TM^ MVN Analyze system in whole-body configuration was used for motion capture; 17 inertial sensors were distributed over the head, torso, arms, hands, legs, and feet. The angles for each coordinate were recorded at 100 Hz frequency using the Xsens own software.

Both systems were synchronized using the Noraxon Myosync channel, which received the start-stop pulses from the Xsens system.

The tested device was the commercial passive lumbar Laevo^TM^ V2 exoskeleton (Figure 2) [19,22,23].

Its objective is relief of back pressure, by helping the user while working in bending, forward, or lifting posture, supporting part of the user’s body weight, reducing stress on the back, and improving the user’s awareness of their posture. According to the manufacturer’s instructions, the size of the exoskeleton was adapted to the anatomy of each participant, and they all were given help putting on the exoskeleton. The weight of the exoskeleton used in the tests is 2.8 kg.

The experiment was led by a technician, who was in charge of instrumenting the inertial sensors and guiding the volunteers in the task development, signal acquisition, and surveillance of the test. A clinical evaluator was also present to carry out EMG sensor placement.

### 3.3. Data Analysis

#### 3.3.1. Assessment of Muscle Activation and Fatigue

Muscle activation and fatigue were analyzed by using EMG signals in four chosen muscles. After the EMG signals were acquired, pre-processing was carried out with a zero-phase bandpass Butterworth filter of order 10. The cut-off frequencies were 20 and 200 Hz to remove movement artefacts and limit the study bandwidth. Muscle activation and fatigue were assessed by calculating certain EMG signal parameters. The first step was to divide the signals to define 16 segments corresponding to the lifting task of each box. The flow chart of the signal treatment process is included in Figure 3. The Figure 4 shows 4 acquired signals during the last 4 movements of a complete exercise (subject 4, exercise with boxes 13–16, without exoskeleton). The marks at the beginning and end of each movement correspond to the fragment where the participant is lifting the box. Marks were located exactly for each case using the MoCap simultaneously recorded by visual inspection. However, these marks only indicate the beginning and end of each movement, not the beginning and end of the myoelectrical activation of the muscle that precedes the mechanical activation. For this reason, marks were relocated one second before, as shown in Figure 4 [56,57].

Then, a parameter is calculated that simultaneously represents the intensity of the four EMG signals. It consists of the sum of the squares of the VRMS values of the four EMG signals, calculated in 0.1 s segments and normalised by the total value of the VRMS of the signal, in order to prevent the amplitude of a signal from influencing the segmentation system. The marks set with the MoCap system, previously, slice the signal and calculate the mean value of all the values that are above 85%, to set a threshold level of 15% of that value. Then, the system places the new start and end marks for the segment.

Finally, a posterior visual check with slight manual correction was carried out, setting the definitive start and end marks.

Six parameters were calculated for each of the 16 segments (movements):Root mean square of the segment (VRMS) (μV).
(1)$$\mathrm{VRMS}=\sqrt{\frac{1}{N}\Sigma_{i=1}^{N}{\left(x[i]\right)}^{2}},$$
where x[i] from i=1 to *N* is the signal segment being analyzed.Zero-crossing rate (TZC, %), relative to the total amount of data in the segment, which provides indirect information of the signal frequency.
(2)$$\mathrm{TZC}=\frac{\Sigma_{i=1}^{N}\left|sgn\left(x\left[i\right]\right)-sgn\left(x\left[i-1\right]\right)\right|}{N}\;\cdot \;100,$$
where sgn(x[i]) from i=1 to *N* is the sign of the signal being analyzed. The following frequency parameters are obtained from the spectrogram (0.5 s window size, non-overlapping). If SP[j,k] is the spectrogram time frequency distribution, where *j* is the time index (from j=1 to *L*, depending on the segment length) and *k* is the frequency index (from k=1 to *M*; M=1024 corresponding to 1 kHz). Then:Mean frequency (FMN, Hz), calculated as the average of the mean power frequency from the spectrogram (0.5 s window size):
(3)$$\mathrm{FMN}=\frac{\Sigma_{j=1}^{L}{\mathrm{FMN}}_{j}}{N},$$
where ${\mathrm{FMN}}_{j}$ is the mean power frequency calculated from time interval of the spectrogram (${SP}_{j}\left[k\right]=SP\left[j,k\right]$), that is:
(4)$${\mathrm{FMN}}_{j}=\frac{\Sigma_{k=1}^{M}k\;\cdot \;SP_{j}\left[k\right]}{\Sigma_{k=1}^{M}SP_{j}\left[k\right]}\;\cdot \;\frac{f_{m}}{M},$$Median frequency (FMD, Hz) calculated as the average of the median power frequency from the spectrogram (0.5 s window size):
(5)$$\mathrm{FMD}=\frac{\Sigma_{j=1}^{L}{\mathrm{FMD}}_{j}}{N},$$
where ${\mathrm{FMD}}_{j}$ is the median power frequency calculated from each time interval of the spectrogram (${SP}_{j}\left[k\right]=SP\left[j,k\right]$), that is:
(6)$${\mathrm{FMD}}_{j}=D\;\cdot \;\frac{f_{m}}{N}\overset{\mathrm{which}\;\mathrm{fulfills}}{\underset{\;}{\to }}\sum_{k=1}^{D}SP_{j}\left[k\right]=\sum_{k=D}^{M}SP_{j}\left[k\right],$$Logarithm of the Dimitrov index ($log({FI}_{n})$) normalized by the minimum value of each exercise and participant, obtained from the spectral marginal of the spectrogram (PSD[k]):
(7)$$PSD\left[k\right]=\sum_{j=1}^{L}SP\left[j,k\right],$$

The Dimitrov index [36,44] is calculated in the band frequency from 20 Hz (k=41) to 200 Hz (k=410):(8)$$FI=\frac{\Sigma_{k=41}^{410}k^{-1}\;\cdot \;PSD\left[k\right]}{\Sigma_{k=41}^{410}k^{5}\;\cdot \;PSD\left[k\right]}\;\cdot \;{\left(\frac{f_{m}}{M}\right)}^{-6}$$

The logarithm is calculated after normalization by the minimum value of each exercise and participant (${FI}_{min}$):(9)$$log\left(FI_{n}\right)=log\left(\frac{FI}{FI_{min}}\right)$$

The objective of the statistical analysis was to find the relationship between the use of the exoskeleton and the calculated variables related to muscular effort and fatigue throughout each task. The factors defining each fragment of signal are the user, the monitored muscle, the position of the box-moving activity, and the weight of the box.

Taking these into account, the mixed model was built as follows:(10)y(var,muscle)∼exo∗box+weight+(1|user)
where exo refers to the condition (with or without exoskeleton), box to the position of the box from one to 16, weight to the box’s weight of 7, 8, or 9 kg, and user appears as the random effect. The mixed model was built using the *lmer* function of the R package lme4 [58]. It was performed for each variable (var): VRMS, TZC, FMN, FMD, and $log({FI}_{n})$ and each muscle: lumbar, gluteus, quadriceps, and semitendinosus. user was included as a random effect after checking the inter-rater intra-class correlation coefficient [59]. The variance introduced by the user was close to the 50% in almost all variables and muscles. A post-hoc analysis was carried out to evaluate the marginal means of the factor exo, to observe the differences between the conditions with and without exoskeleton. It also was performed to observe these differences by box position, that is the factor box. The model adjustment is done using the function *interactionMeans* of the R phia package [60,61]. This function calculates the adjusted means of the fitted mixed model, plus the standard error of those values, for all the interactions of given factors. The methodology used was the Holm methodology [62]. All calculations were done with R, using the pwr package [63] for the normality test, lme4 [58] for ANOVA, and phia [60] for the post-hoc analysis.

The results are shown in the next section; only the variables showing significant differences are included. The results are given as percentage of reduction with respect to the condition without the exoskeleton, as follows:(11)$${\%}_{var}=\frac{Mean_{var,exo}-Mean_{var,noexo}}{Mean_{var,noexo}}$$

A negative value of this percentage for a certain parameter means a reduction with respect to the condition without exoskeleton, and a positive value means an increase.

Also, in Appendix A it is included a set of plots for each variable and muscle mixed model fitting. These plots represent the residues (fitted values minus observed values) over the fitted values.

In Appendix B it is shown a series of tables that contain the extended information of the studied model. Similarly, as stated in Equation (Equation 11), the percentage of reduction are calculated, but adding the fixed factor box. So, the reductions can be observed by condition and by box position.

#### 3.3.2. Assessment of Posture

The influence of the exoskeleton on posture was evaluated with the use of a motion capture (MoCap) system. The five percentiles ($P_{5}$, $P_{25}$, $P_{50}$, $P_{75}$, and $P_{95}$) of the trunk and hip were obtained from the MoCap.

The range of motion was calculated as the difference between extreme percentiles:(12)$$P_{RoM}=P_{95}-P_{5}$$

A *t*-test was performed in R to find significant differences (*p*-value < 0.05) between the two conditions.

## 4. Results

### 4.1. EMG

#### 4.1.1. Muscle Activation (VRMS)

Significant differences were found for this variable for all four muscles, and in all cases the difference meant reduced VRMS when wearing the exoskeleton (Table 1).

In the case of the erector spinae, the reduction was 8%. The VRMS graph shows a positive slope (Figure 5a). This trend indicates that muscular activity increased at a quite constant proportion throughout the exercise. Experimentally, in terms of average VRMS for all boxes, the quadriceps shows a decrease of 10.2% in the exoskeleton condition (Table 1). Figure 5c shows there is a positive overall trend for both curves (exo and no exo). The values in both conditions overlap for some box handling, but are clearly higher with the exoskeleton for, e.g., box 14, located at the floor level; however, this effect is not observed in the rest of the boxes.

As shown in Table 1, the VRMS for gluteus shows a 5% reduction when wearing the exoskeleton. It can be observed in Figure 5b that while the trend is positive, similar to the trend for the erector spinae, the biggest difference between the exoskeleton and no exoskeleton condition is found with the last boxes. These are the boxes located closest to the floor, thus demand the maximum effort for this muscle; also, the effective value, represented by VRMS, is higher.

The 14% VRMS reduction for the semitendinosus muscle (Table 1) shows that this muscle received the most benefit from the exoskeleton. As shown in Figure 5d, this release is approximately constant throughout the exercise, along with the VRMS trend during the series, which is not have positive, but flattened, with soft peaks for every fourth box of each row (4, 8, 12, and 16).

#### 4.1.2. Linear Fatigue Parameters: TZC, FMN and FMD

An increase in the values of TZC, FMN and FMD, showing a positive percentage, implies a decrease in fatigue with the exoskeleton. So, contrary to the VRMS results, a positive percentage, or an increase, means a reduction in fatigue. The only muscle showing a significant increase in TZC, FMN, and FMD is the semitendinosus, with 1.7, 0.9, and 1.12%, respectively. As described in the previous subsection, the semitendinosus was observed to have the most reduced muscle activity (VRMS). As the numbers show, this also indicates a decrease in fatigue (Table 1).

In the case of erector spinae and gluteus, the trend of the curves of each parameter and muscle follows a similar negative pattern in almost all cases (Figure 6, Figure 7 and Figure 8a,b). This proves that these muscles suffered increased fatigue as the subject continued to carry the series of boxes. Nevertheless, no differences were observed in the process of fatigue between the two conditions (with and without exoskeleton), nor was a different trend observed. The semitendinosus, however, does show differences, as seen in the results in Table 1 and Figure 6, Figure 7 and Figure 8d. In all three graphs, it can be observed that the curves for the exoskeleton condition, corresponding to the legend label “With Exo”, are bellow the “Without Exo” curves, specially in the last boxes. Meaning a slowdown in the process of fatigue. In the case of the quadriceps, the muscle that is not assisted by the device, the curve is not constant; there is no clear trend (Figure 6, Figure 7 and Figure 8c). The behavior and motions observed with the last four boxes showed decreased fatigue when wearing the exoskeleton.

#### 4.1.3. Non-Linear Fatigue Parameters: Log(FImin)

With regard to the normalized Dimitrov’s index, for the muscles that potentially benefit from the exoskeleton (erector spinae, gluteus, and semitendinosus) significant differences were found (Table 1). These differences correspond to −11.5% for the erector spinae, −14% for the gluteus, and −8% for the semitendinosus. For this parameter, a decreased percentage implies a decrease in fatigue (unlike the previous parameters). These reductions imply that all three mentioned muscles show reduced fatigue with the use of the exoskeleton.

The trend observed is positive, meaning the muscles suffer increased fatigue as the subject continued carrying the series of boxes, which agrees with the rest of the fatigue parameters (Figure 9a–d). In all cases, the curve related to the condition with the exoskeleton is above the curve for the condition without, implying a diminution in fatigue when wearing the assistance device. In the case of the quadriceps, the trend is irregular and the effect of the device on the fatigue process is uneven (Figure 9c).

### 4.2. Motion Capture

The results of the motion analysis in the present study demonstrate significant differences in some of the joint coordinate ranges and percentiles (Table 2). In almost all the cases, the differences were smaller than 10%. In the case of the lower back range of motion, in extension and rotation, we could observe reductions of 3% (*p*-value = 0.001), and 39% (*p*-value = 0.01), respectively. No other differences in range of motion were found; the $P_{5}$ showed an 8% reduction in right hip rotation and a 5% reduction in right knee flexion.

## 5. Discussion

The lumbar muscles are, a priori, the main beneficiaries of the exoskeleton. The results show a decrease in muscle activity and a decrease in fatigue expressed in terms of Log(FImin). When comparing our results obtained with those of studies mentioned in the Introduction, agreement is found with the outcomes obtained by Antwi-Afari et al. [27] and Di Natali et al. [28], who also observed reduced lumbar muscle activity when monitoring people performing load-lifting tasks. These are examples of the many studies that measured the back muscles and observed reductions [64,65] when assessing their own or a third-party designed exoskeleton.

Some authors assessing the same model of exoskeleton, the exoskeleton Laevo ^TM^ V2 also found significant reductions in the back muscles, 11–57% [22] and 35–38% [35]. On the other hand, the authors Amandels et al. [19] found no differences for erector spinae. Comparing the results obtained regarding the fatigue, the present results agree with the obtained by the authors Lotz et al. [40] and Godwin et al. [41], who also calculated the median frequency to evaluate the fatigue of a lumbar exoskeleton and found a reduction of fatigue in the lumbar spinae. Other authors also found reduction of fatigue when using a lumbar exoskeleton, but since the parameters used to evaluate were different, no specific comparison was carried out [35,42].

In the designed task of the present study, muscular activity increased in a quite constant proportion throughout the exercise. The exoskeleton had a more evident effect in moments when the demand of muscle activity was higher. The quadriceps was chosen as the potentially prejudiced muscle, for compensating the reduced activity of lumbar muscles. It participates in flexion of the hip, and the exoskeleton is designed to discharge the lumbar flexion and to charge it on the hip flexion. Like the rest of the muscles, on balance, this muscle also showed reduced activity, although with the smallest significance (*). Contrary to the other muscles, no significant change in terms of fatigue was observed. Hence, the hypothesis that there would be a negative effect on the quadriceps was not supported. Glinski et al. [66] assessed a lumbar exoskeleton with comparable performance to the Laevo, with leg pads located over the quadriceps area. In that study, they observed an increase in right quadriceps activation, and a decrease in left quadriceps activation. In the present work, the left leg was analyzed leg, because of the asymmetrical configuration of the task, which had a bigger demand on this side. In this case, the result agrees with the cited work when comparing left quadriceps outcomes, but no conclusions can be drawn for the right quadriceps. The gluteus and semitendinosus function by supporting the extension of the hip joint, and theoretically, by using the exoskeleton, which facilitates this motion, they should manifest relief. Both the gluteus and semitendinosus presented reduced activity with the exoskeleton.

In terms of fatigue, both muscles showed a reduction as expressed by the Log(FImin), and the semitendinosus by TZC, FMD, and FMN. However, studies including the semitendinosus and gluteus focused on walk-assisted exoskeletons [67,68,69], and no comparisons were made. These were considered relevant in this study because passive systems (and in general systems that are not anchored to the ground) can only work by modifying the load conditions between different body segments. Since the lumbar exoskeleton basically covers the lumbar, sacral, and hip joints, it can modify the loading conditions between these segments, and thus modify the activation of muscle groups that control these joints. The gluteus and semitendinosus, together with the quadriceps, are the most powerful muscle groups that are involved, although not solely, in the control of the hip joint. The results of the gluteus and semitendinosus are consistent with the theoretical expectation of exoskeleton performance. Although in the present work the computation of traditional temporal and spectral parameters has been considered to assess the effect of the exoskeleton on muscle fatigue, since they all point in the same direction, literature proposes the use of other time-frequency or complexity sEMG parameters for the assessment of muscle fatigue that could be considered in future works [70].

The results of motion analysis in the present study show reductions the movement of the back joints as a consequence of slight restriction imposed by the exoskeleton. While the differences in the joints related to the muscles studied were small, they reveal a change in strategy for the tasks as a result of using the device. This observation is in agreement with the review of Pesenti et al. [26]. A higher difference is observed in the lumbar rotation coordinate, however, this motion is not related to the studied muscular activity, for none of the chosen muscles is involved in the lumbar rotation. This is one of the limitations of the present study, in a further assessment, the muscles: external oblique, rectus abdominis, or the lumbar multifidus should be included. In addition, with the gathered motion data, a future study could be carried out to include a deeper kinematic analysis. The ranges of motion could be analyzed using an analogue segmentation as the one performed in the present study. With this approach, the results of kinematics and EMG could be combined by obtaining mixed indexes to characterize the effects of the exoskeleton to a greater extent. To evaluate further passive exoskeletons (for example possible mixed performance indexes that take into account both EMG and Kinematics) could be added.

This work analyzed the use of a passive lumbar exoskeleton in laboratory conditions. The experimental results show that the use of an exoskeleton led to a significant reduction in the effective value of muscular activity in all analyzed muscles. In addition, a decrease in fatigue was observed in the erector spinae, semitendinosus, and gluteus muscles, as indicated by a series of parameters. No differences were found in the fatigue process of quadriceps, the muscle considered to be potentially adversely affected. The curves plotted for each muscle throughout the exercise with the boxes clearly showed a tendency of increased effective value and fatigue as the exercise proceeded.

Despite the risk of the exoskeleton constraining natural movement, in this study, only small percentage reductions in range of motion were found in the lumbar joint, and the modifications in the participants’ mobility patterns did not significantly affect the fatigue process. Some other coordinates, such as in the right hip and knee, showed a reduction in the 5th percentile of range of movement. This may be associated with the change in motion strategy of participants when handling the weights while wearing the exoskeleton.

## 6. Conclusions

The results suggest that the use of a lumbar exoskeleton provides a benefit to the worker, taking into account the muscles that can potentially benefit, but also those for which are expected get negative impact. Despite the observed advantages of an exoskeleton, there is still plenty of room for improvement. The repercussion of the restrictive effects in motion could be further analyzed. Also, it must be borne in mind that the study scope was the objective evaluation of tasks with short duration and limited type of movements; for further evaluation of acceptance and long-term effects, a longitudinal study, with the inclusion of more variety of tasks, will be carried out. Companies must take into account that the use of this equipment may be a possible solution in cases where other technical or organizational measures are not feasible or effective for reducing the physical load in the workplace. An ergonomic evaluation and a redesign of the job considering the results of such evaluation should always be the first way to improve a job, considering other measures such as the use of exoskeletons when efforts are exhausted and the expected improvement has not been achieved.

## Figures and Tables

**Figure 1 sensors-22-04060-f001:**
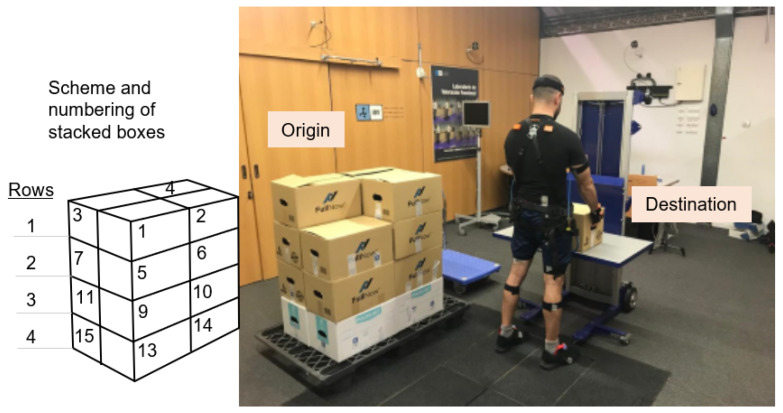
(**Right**): Picture of test set up. Structure of the pallet of 16 boxes and participant fully instrumented with exoskeleton, EMG sensors and MoCap inertial sensors. Origin: where boxes on pallet are picked up. Destination: table where user places boxes. (**Left**): Illustration of setup scheme. Numbers represent the order boxes are picked up.

**Figure 2 sensors-22-04060-f002:**
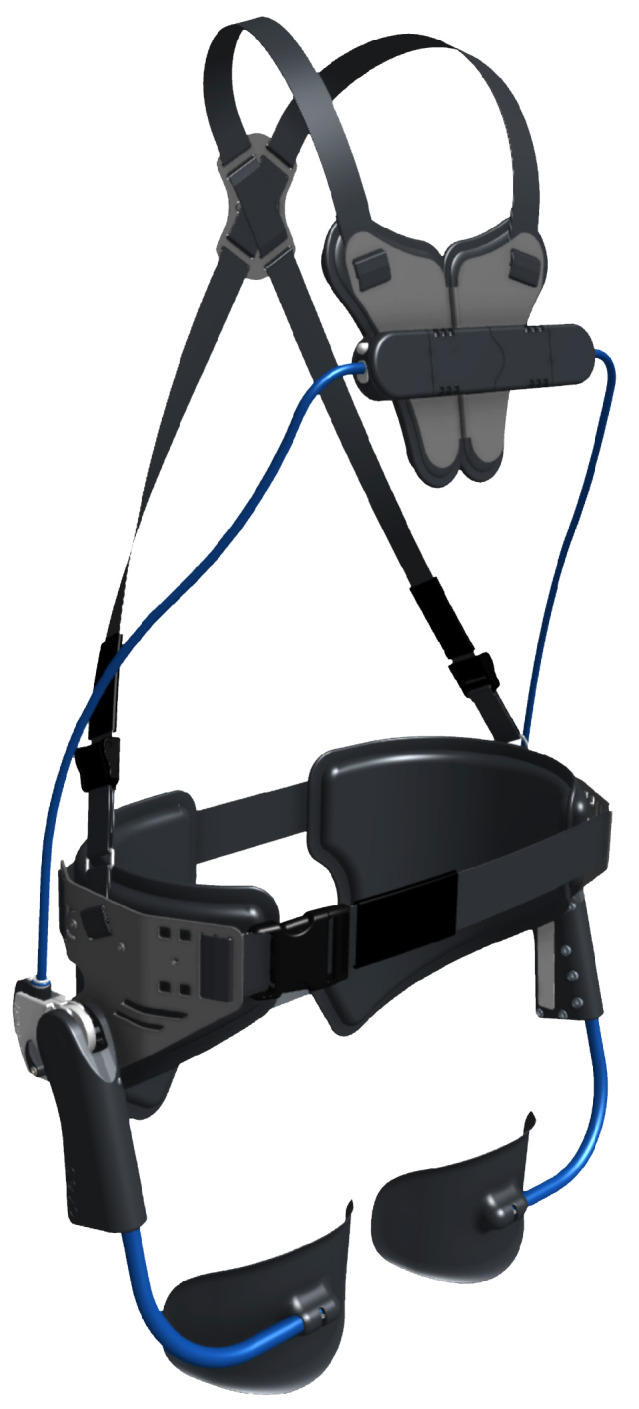
Assessed lumbar Laevo^TM^ V2 exoskeleton.

**Figure 3 sensors-22-04060-f003:**
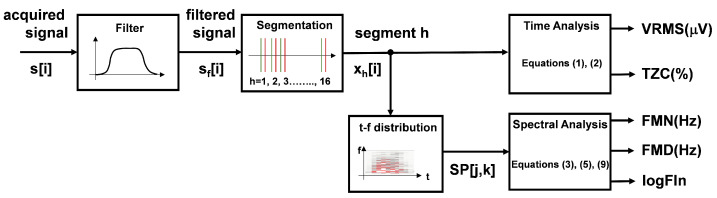
Flow chart of the EMG signals processing. Filtering, segmentation, and parametrization.

**Figure 4 sensors-22-04060-f004:**
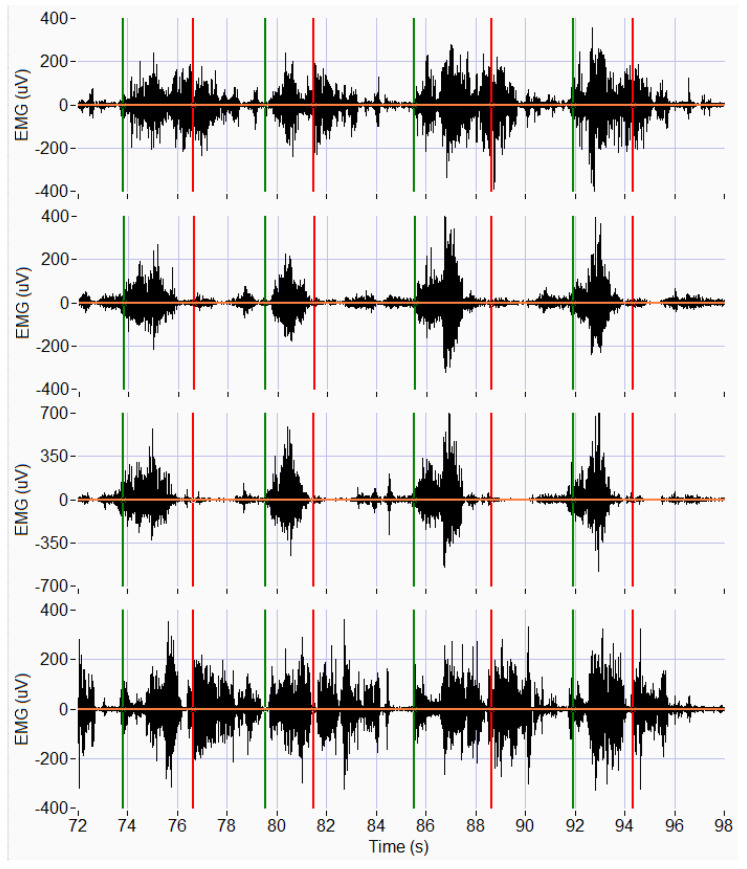
Surface EMG signals acquired simultaneously from four muscles. From top to bottom: erector spinae, gluteus medius, quadriceps femoris, semitendinosus. Time window corresponds to last 4 movements of exercise by subject 4 without exoskeleton. Vertical marks indicate beginning (green) and end (red) of myoelectrical activation, before visual check carried out to catch the boxes.

**Figure 5 sensors-22-04060-f005:**
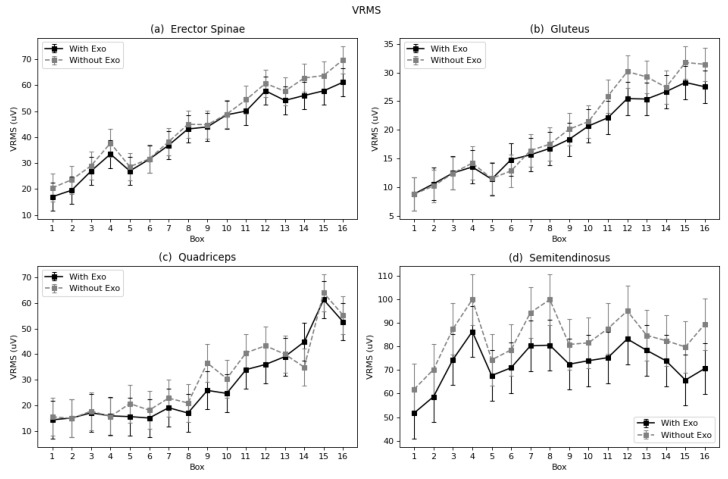
Marginal mean curves with the standard error bars of VRMS parameter throughout the 16 boxes for all four muscles: (**a**) Erector Spinae, (**b**) Gluteus, (**c**) Quadriceps, and (**d**) Semitendinosus.

**Figure 6 sensors-22-04060-f006:**
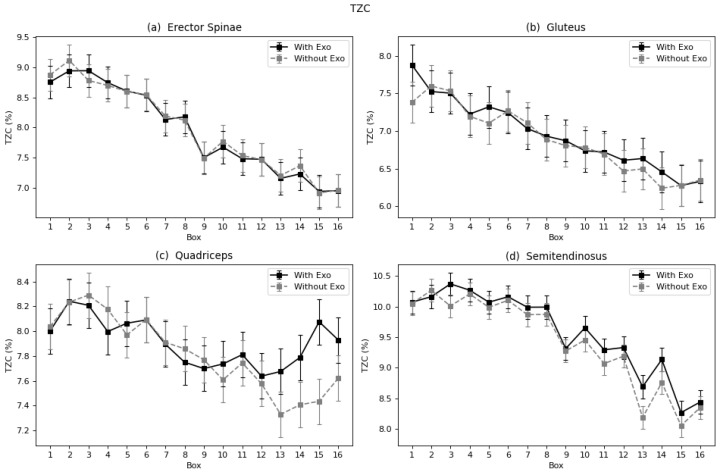
Marginal mean curves with the standard error bars of TZC parameter throughout the 16 boxes for all four muscles: (**a**) Erector Spinae, (**b**) Gluteus, (**c**) Quadriceps, and (**d**) Semitendinosus.

**Figure 7 sensors-22-04060-f007:**
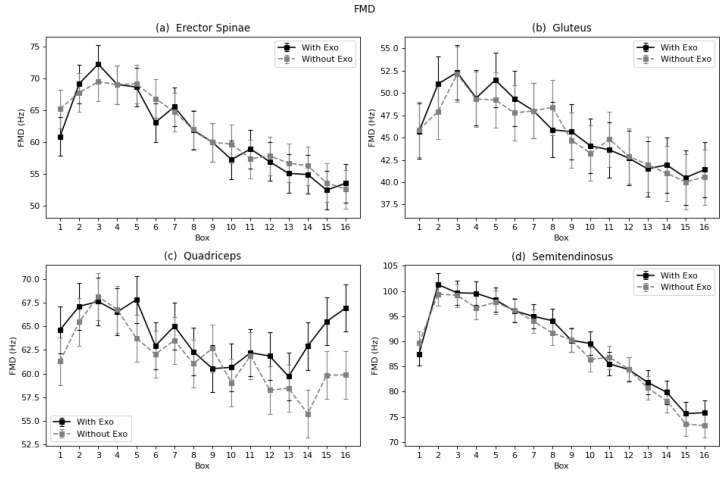
Marginal mean curves with the standard error bars of FMD parameter throughout the 16 boxes for all four muscles: (**a**) Erector Spinae, (**b**) Gluteus, (**c**) Quadriceps, and (**d**) Semitendinosus.

**Figure 8 sensors-22-04060-f008:**
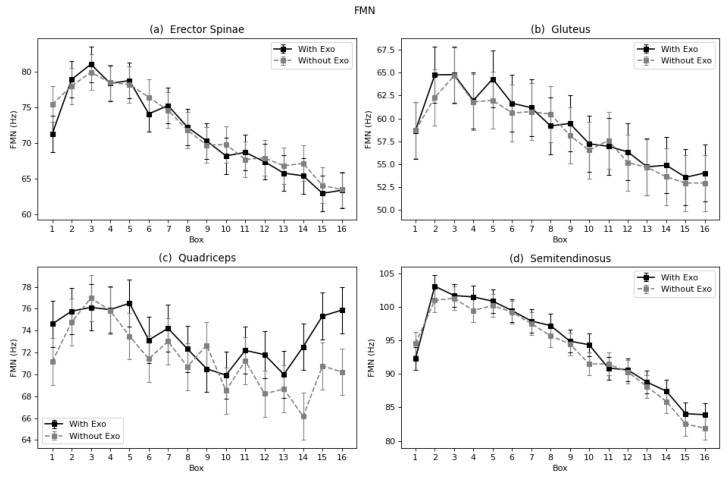
Marginal mean curves with the standard error bars of FMN parameter throughout the 16 boxes for all four muscles: (**a**) Erector Spinae, (**b**) Gluteus, (**c**) Quadriceps, and (**d**) Semitendinosus.

**Figure 9 sensors-22-04060-f009:**
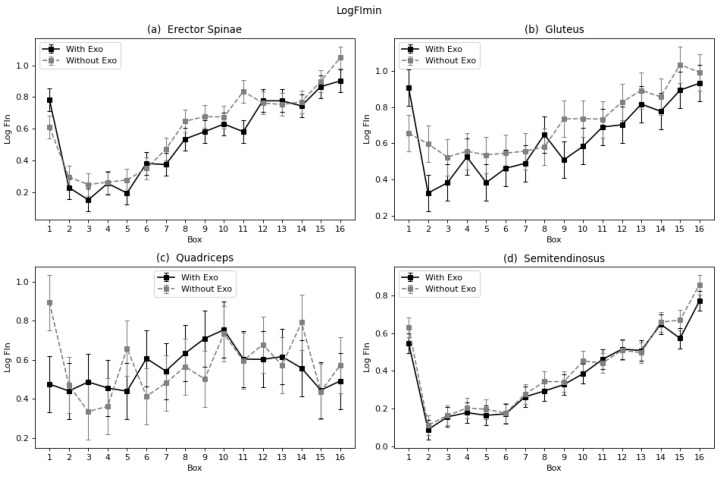
Marginal mean curves with the standard error bars of Log(FImin) parameter throughout the 16 boxes for all four muscles: (**a**) Erector Spinae, (**b**) Gluteus, (**c**) Quadriceps, and (**d**) Semitendinosus.

**Table 1 sensors-22-04060-t001:** Percentage of variation in EMG variables caused by exoskeleton, together with their confidence interval for a sample size of 768 and a level of confidence of 95%. Values with significant differences are show in bold, with *p*-values below 0.05. *: *p*≤ 0.05, **: *p*≤0.01, ***: *p*≤ 0.001,****: *p*≤ 0.0001.

Muscle	VRMS(%)	TZC(%)	FMN(%)	FMD(%)	log(FIn)(%)
Erect.S.	−8±3 ****	−0.3 ± 0.9	−0.6 ± 0.9	−0.8±1.4	−11±6 **
Semit.	−14±2 ****	1.7±0.9 ***	0.9±0.7 *	1.12±1.08 *	−8±7 *
Glut.	−5±4 **	1.1 ± 1.3	1.4 ± 1.5	1 ± 2	**−14** ± **9** **
Quad.	**−4** ± **4** *	0.5 ± 1.1	0.2 ± 1.1	0.4 ± 1.8	4 ± 11

**Table 2 sensors-22-04060-t002:** Percentage reduction in joint percentiles and range caused by the exoskeleton with their interval of confidence. The table shows the figures where the *p*-values are below 0.05.

Lumbar %		Right Hip %		Right Knee %	
$P_{RoM}$ Flexion Extension	−3 ± 1	$P_{5}$ Rotation	−8 ± 5	$P_{5}$ Flexion Extension	−5 ± 3
$P_{RoM}$ Rotation	−39 ± 14				

## Data Availability

Not applicable.

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
