# Peer review of "Assessment of a Passive Lumbar Exoskeleton in Material Manual Handling Tasks under Laboratory Conditions"

_sensors, 2022, doi:10.3390/s22114060_

Round 1

Reviewer 1 Report

This article presents a study on the effect of using a passive lumbar exoskeleton in terms of moderate ergonomic risk, and it indicates that the passive exoskeleton reduces muscle activity and introduces a minor change of strategies for motion. However, the article lacks theoretical and technical innovation, and it looks more like an experimental report than a research article. I think that this work needs to further explore the impact of exoskeletons on human muscles during labor, such as assisting effect in different handling postures, and discussing how to improve the design of exoskeletons to achieve better results. Some specific suggestions for this article are listed below:

  1. What does the sentence “the differences were smaller than 10” mean in the abstract, and what is the unit of 10?
  2. It is obvious that exoskeletons can reduce muscle fatigue, so is this paper just to verify this obvious result? The author should give a sufficiently attractive conclusion and purpose in the abstract.
  3. The beginning of Section 2 has two completely repeated paragraphs, and the author should double-check to avoid such a low-level mistake again.
  4. The test scenarios and actions shown in Figure 1 are too simplistic. In fact, the human body uses a variety of different postures in the process of carrying heavy objects, and the activity of the muscle under different postures is very different. This is also one of the problems that the current waist exoskeleton has not solved.
  5. FIG. 3 can preferably provide a flow chart of the processing of the EMG signal, and a comparison diagram of the EMG signal before and after processing.
  6. The author uses EMG equipment to collect EMG signals, and uses motion capture system to collect human motion postures. Is it necessary to ensure the synchronization of the two signals in time, and how to ensure the synchronization of signals from different systems?

Author Response

The lines in the comments are referenced to the document with the changes made explicit.

Reviewer 2 Report

This paper provides an experimental evaluation of a commercial lumbar
exoskeleton via analysis of electromyographic signals. It is well written and
organized, and provides useful information to the community.

I recommend the following minor suggestions aimed at enhancing the paper:

1. Add detail about the experimental protocol:
  - which order were the conditions (exo on/off) presented to each participants
  - how might this effect the outcomes (address in the discussion)
  - what was the frequency of the metronome
  - if only 4 emg channels were used, were the electrodes placed on one
    side of the body (e.g., left or right) of the participants body? how was
    the side chosen?

2. Add details or clarify the spectral analysis.
  - [line 209] what does "0.5 seconds time steps" refer to? is this the window
    size or something different?

3. Enhance the description and reporting of the linear mixed-effects model:
  - why not report the parameters and confidence intervals associated with Eq.
    10? At least for the 'exo' regressor for the significant outcomes? Or,
    provide confidence intervals for the estimates in Table 1 (if this is
    possible). 
  - A plot demonstrating the overall model fit, e.g., fitted vs observed
    values?
  - Does the random effects (e.g., 'user') explain a large portion of the
    variance? (e.g., variance partition coefficients or intraclass
    correlations). Does the users' ability have an effect on overall
    performance (address in the discussion)?
  - add a brief description of how [line 238] "evaluate the marginal averages"
    was accomplished, e.g., explain what "marginal mean curves" means
  - add a description of what information the error bars denote in each of the
    Figs.4-8

4. Some clarification on the "Linear fatigue parameters".
  - it's not clear what the authors are suggesting in
  [lines 294-296] "...have smaller slope"
  To me the lines look nearly the same for all three plots, and only have minor
  differences at certain boxes. it might be helpful to identify where the
  slopes are "smaller" for the reader.

5. A moderate amount of typos or minor clarifications needed:

[line 12] "were smaller than 10"  - what are the units?

[line 21] extra "39%"
[line 52] "HR" - what is the acronym mean?
[lines 58 and 59] "was" -> were
[line 77] "fulfil" -> fulfill
[line 106] "IMC" - what is the acronym?
[lines 114-122] redundant information
[line 124] "was" -> were
[caption Fig.1] "inertials" -> inertial sensors
[line 238] "post-hot" -> post-hoc
[line 248] "TThe" -> The
[Section 3.1.1] The paragraph format feels a bit odd in this section.
[Figs. 4-8] What is the meaning of "ergo-areas"?
[line 281] "Lineal" -> linear
[line 316] "10" - what are the units of this measure?
[line 332-333] I don't understand the sentence: "It participates in flexion of
the hip, and the exoskeleton is designed to discharge the erector spinae to
charge it on this muscle." - please clarify
[line 340] "left leg..." - this information could be provided in the methods
section, with a bit of clarification why the left leg was chosen (see comment
1)
[lines 355-357] numbers in text should be removed

Author Response

(The authors gave the same response as above.)

Reviewer 3 Report

File attached.

Author Response

(The authors gave the same response as above.)

Round 2

Reviewer 1 Report

As I mentioned before, the test scenarios and actions are too simplistic. I think the authors need to add tests and analyses about humans using more different postures for carrying heavy objects.

Author Response

Please find the videos in the supplementary material.

Author Response

Find the response in the attached file

Round 3

Reviewer 1 Report

Although the author did not revise it according to my suggestion, I still gave it a pass. I hope the author can continue to do more in-depth research.

Author Response

Thank you very much for all your comments throughout the three rounds. We are very sorry that we have not been able to revise according your suggestion.

As we responded in last round, the whole experimental protocol design, and the conception of this study was made following a series of criteria backed by other works found in literature. Our main purpose was finding physiological evidences of the impact in muscle fatigue regarding the use of the exoskeleton. This requires a controlled design of experiments. Increasing the variability of movements will hide the impact of the exoskeleton in the variance of data. From our point of view, this type of research is worthy, but requires a different type of tools, and it is better faced from the perspective of users of the exoskeletons.

In fact, it could help to answer further questions like in which degree of complexity, or in which kind of movements, time of use, etc. the good performance and effects of the exoskeleton is still as it showed to be in the tasks described in the paper. This way, additionally to get the feedback from tasks like the one presented, a depalletizing task, we could have a bigger picture of the exoskeleton application to workstations. It is true that, as we mention in our
past answer, some authors found the exos not to be appropriate for complex tasks, but could be interesting to find a middle point, and to better characterize the workstations for which the exoskeleton is suitable.

Changes in the manuscript:

In the conclusions, explicit mention to the fact that the motion was limited and that the future research should take that into account, is included p.16-17, 554-557: “Also, it must be beared in mind that the study scope was the objective evaluation of tasks with limited duration and limited type of movements; for further evaluation of acceptance and long-term effects, a longitudinal study, with the inclusion of more variety of tasks, should will be carried out as the next step.”

Reviewer 3 Report

Thanks to the authors for answering my comments.

Some minor modifications to improve readability:

- maybe transfer the detailed review on passive exoskeletons from the introduction to the state of the art: the state of the art section could become

2.1 Passive exoskeletons and common evaluation : Review on passiv exoskeletons and the common evaluation methods (EMG amplitude,...)

2.2 Reduction of fatigue assessment (current 2.1, the current title chosen by the authors for this subsection is not very clear)

2.3 Spectral analysis (current 2.2, there is a typo in the title of this 2.2 section, "evaluation" can be removed)

This would allow to shorten a bit the introduction to improve readability.

- line 363, p.10 : put "with or without the exoskeleton" between parenthesis

- In the conclusion or discussion, perspectives about a deeper kinematic analysis to evaluate further passive exoskeletons (for example possible mixed performance indexes that take into account both EMG and Kinematics) could be added
